

# Modification, Calibration, and Performance of the Ultra-High Sensitivity Aerosol Spectrometer for Particle Size Distribution and Volatility Measurements During the Atmospheric Tomography (ATom) Airborne Campaign

Agnieszka Kupc[1,2], Christina Williamson[1,2], Nicholas L. Wagner[1,2], Mathews Richardson[1,2], Charles A. Brock[1]

[1]Chemical Sciences Division, Earth System Research Laboratory, National Oceanic and Atmospheric Administration, Boulder, CO, 80305-3337, USA
[2]Cooperative Institute for Research in Environmental Sciences, University of Colorado, Boulder, CO 80309, USA

*Correspondence to*: Agnieszka Kupc (agnieszka.kupc@noaa.gov)

**Abstract.** Atmospheric aerosol is a key component of the chemistry and climate of the Earth's atmosphere. Accurate measurement of the concentration of atmospheric particles as a function of their size is fundamental to investigations of particle

microphysics, optical characteristics, and chemical processes. We describe the modification, calibration, and performance of two commercially available, ultra-high sensitivity aerosol spectrometers (UHSASs) as used on the NASA DC-8 aircraft during the Atmospheric Tomography Mission (ATom). To avoid sample flow issues related to pressure variations during aircraft altitude changes, we installed a laminar flow meter on each instrument to measure sample flow directly at the inlet as well as flow controllers to maintain constant volumetric sheath flows. In addition, we added a compact thermodenuder operating at

300˚C to the inlet line of one of the instruments. With these modifications, the instruments are capable of making accurate (ranging from 7 % for $D_p$ <0.07 µm to 1 % for $D_p$ > 0.13 µm), precise (< ±1.2 %) and continuous (1 Hz) measurements of size-resolved particle number concentration over the diameter range of 0.063-1.0 µm at ambient pressures of >1000 to 225 hPa, while simultaneously providing information on particle volatility.

We assessed the effect of uncertainty in the refractive index ($n$) of ambient particles that are sized by the UHSAS assuming

the refractive index of ammonium sulfate ($n$=1.52). For calibration particles with $n$ between 1.44 and 1.58, the UHSAS diameter varies by +4/-10 % relative to ammonium sulfate. This diameter uncertainty associated with the range of refractive indices (i.e. particle composition) translates to aerosol surface area and volume uncertainties of +8.4/-17.8 % and +12.4/-27.5 % respectively. Additional to sizing uncertainty, low counting statistics can lead to uncertainties of <20 % for aerosol surface area and <30 % for volume with 10 s time resolution. The UHSAS reduction in counting efficiency was corrected for

concentrations >1000 cm[-3].

Examples of thermodenuded and non-thermodenuded aerosol number and volume size distributions as well as propagated uncertainties are shown for several cases encountered during the ATom project. Uncertainties in particle number concentration





were limited by counting statistics, especially in the tropical upper troposphere where accumulation mode concentrations were sometimes <20 cm$^{-3}$ (counting rates ~5 Hz) at standard temperature and pressure.

## 1. Introduction

The concentration of particles as a function of size is fundamentally related to both direct (aerosol-radiation) and indirect
(aerosol-cloud) effects of aerosol on climate. Particles with diameters ($D_p$) > 0.1 µm efficiently scatter and absorb solar radiation (e.g., Charlson et. al., 1992). Particles with $D_p$ >0.05 µm serve as cloud condensation nuclei (CCN; Clarke and Kapustin, 2002; Dusek et al., 2006; Köhler 1936). CCN play a role in cloud formation and in altering radiative properties and lifetime of existing clouds (Albrecht, 1989; Twomey, 1974, 1977). Measurement of aerosol size-resolved number concentration is crucial to understand aerosol sources and sinks, optical properties, cloud nucleation potential and chemical
transformations, and consequently to constrain models of aerosol-cloud-climate interactions.

There is currently a variety of techniques available for measuring aerosol size distributions (McMurry, 2000) but only some of these are fast enough to sample aboard aircraft. The ultra-high sensitivity aerosol spectrometer (UHSAS; Droplet Measurement Techniques (DMT) Inc., Longmont, CO, USA) is one such instrument. The UHSAS is an optical particle counter for measuring particles from 0.06-1 µm, which is often used for laboratory, ground-based and airborne measurements. It counts
and sizes particles by measuring the amount of light scattered by individual particles as they traverse a focused laser beam. A fraction of the side-scattered light is then collected by the optical system and focused onto two photodetectors where it is converted to a proportional voltage pulse. The size of particle is determined from the height of the voltage pulse by using a calibration curve obtained from measurements of spherical particles with known size and composition. Size distributions are obtained by accumulating the individual pulse magnitudes of a population of particles into a histogram.

Two versions of UHSAS are currently commercially available. One, designed for airborne measurements, is enclosed in an underwing canister for in-situ sampling, while the other one is intended for ground-based aerosol sampling. Here we focus on the modification, accuracy and operation of two UHSAS instruments (hereafter referred to as UHSAS-1 and UHSAS-2) during the first and second ATmospheric Tomography Mission (ATom) field campaigns in summer 2016 and winter 2017, respectively. The ground-type UHSAS instruments were chosen for this study over the wing-mounted version because we
wished to dry the air sample and install a thermodenuder used to distinguish non-volatile particles, these sample treatments are not possible with the compact, wing-mounted instrument. The ground-type UHSAS has been deployed in various airborne-based campaigns (Brock et al., 2011, 2016; Kassianov et al., 2015; Yokelson et al., 2011). However, as reported by Brock et al. (2011), modifications to the flow system are required to make them suitable for airborne sampling.

Cai et al. (2008) reported a laboratory evaluation of the UHSAS and Brock et al. (2011) report modifications to the flow
system; however, a complete evaluation of the accuracy and precision of the UHSAS instrument for airborne operation is lacking. Here we describe modifications to the ground-based UHSAS for airborne operation, detail the installation of a compact



thermodenuder in a second UHSAS for aerosol volatility studies, and evaluate the accuracy, precision and in-flight performance of both UHSAS instruments during the first two of four ATom airborne campaigns.

## 1.1 The ATom mission

The ATom mission uses a DC-8 aircraft to survey the remote atmosphere over the Pacific and Atlantic Oceans from ~80˚ N to ~65˚ S while making repeated vertical profiles from 0.15-12 km to provide information on greenhouse gases, reactive and tracer species, and aerosol composition and size distribution. At the conclusion of the ATom project in spring 2018, the DC-8 will have made four global circuits, one circuit for each season. The UHSAS instruments are a part of a suite of fast-response aerosol size distribution instruments focusing in particular on the spatial variation in the abundance of 0.003-4.8 μm sized particles (Brock et al., in preparation; Williamson et al., in preparation). Scientific goals for these instruments include identifying the spatial extent of new particle formation in the remote troposphere and the associated mechanisms and controlling parameters, quantifying the growth of newly formed particles to cloud-active sizes, and determining the importance of aerosols from continental sources to the remote troposphere.

By operating two well-calibrated UHSAS instruments, one with a thermodenuder (UHSAS-1), and one without (UHSAS-2), the size-dependent particle volatility can be determined continuously, which is particularly useful for airborne sampling where fast time response is needed. Volatility is an important physical property defined by the chemical composition of the condensed species and may reflect the origin of the particle (Huffman et al., 2008; Jonsson et al., 2007). Most secondary compounds (such as sulfates, nitrates or organics) are expected to volatilize below 300˚C while primary particles such as soot, sea salt and soil dust survive heating (e.g., Clarke, 1991; Clarke and Kapustin, 2002; DeCarlo et al., 2008). Measurements of particle volatility help identify the contribution of secondary particles formed in the free troposphere (FT) to the budget of CCN-sized particles in the marine boundary layer (MBL), and how this contribution varies with altitude and location in the remote atmosphere.

## 2. The ultra-high sensitivity aerosol spectrometer (UHSAS)

### 2.1 Operating principles

The UHSAS (Cai et al., 2008) measures aerosol size-resolved number concentration between 0.06-1 μm in 99 logarithmically spaced bins with user-selected time resolution. The UHSAS uses a high-intensity infrared laser (a semiconductor-diode-pumped solid-state neodymium-doped yttrium lithium fluoride ($Nd^{3+}$:Y LiF$_4$), operating at 1054 nm with intra-cavity circulating power of ~1 kW cm$^{-2}$), an inlet jet assembly and two detection systems: a highly sensitive avalanche photodiode (APD) to detect and size the smallest particles, and a less sensitive secondary PIN photodiode to size larger particles (Fig.1). These detectors are located at 90˚ on either side of laser beam, aligned with the intersection of the aerosol stream and the laser beam.



When particles exit the inlet jet assembly in the optical block they traverse the center of the focused laser beam and scatter light into the detection system. Scattered light collected by two pairs of Mangin optics (over solid angle of 33-147˚) is imaged onto the APD and PIN photodetectors. The center region of the solid angle (72.5-104.8˚) is not sampled due to the hole cut out in the outer of the mirrors, with the detector size being a negligible fraction of this hole area (Brock et al., 2016). This geometry

5   contrasts with that reported by Cai et al. (2008), who used scattering angles from 22-158˚ to simulate UHSAS response. The geometry we report was determined in consultation with the manufacturer and agrees with values given by Petzold et al. (2013). The amount of scattered light reaching the detectors is a function of not only particle size, but also refractive index ($n$) and shape.

Each photodiode produces a photocurrent pulse, which is converted to a voltage pulse through analog amplifiers. The signal

10   from each detector is amplified by two different gain circuits (high and low), providing a total of four independent gain stages with some overlap (one particle may be separately sized by each of two adjacent gain stages). The outputs from these gain stages are combined by linear regression in the overlap regions to provide a single scale for accurate sizing across the full range of the UHSAS response.

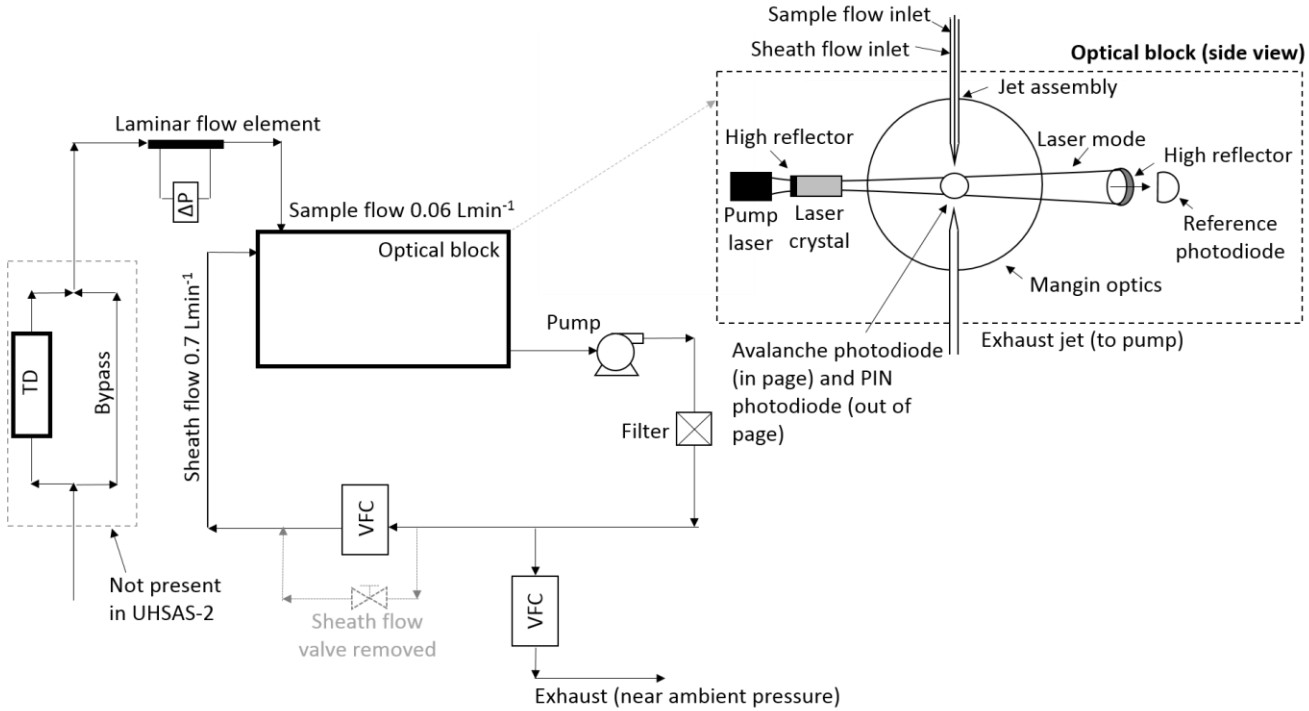

**Fig.1: UHSAS with its modified flow system including schematic of the UHSAS-1 and UHSAS-2 inlets. The sample air at a flow rate of 60 cm³min⁻¹ enters the inlet and in UHSAS-2 goes directly through the laminar flow element, while in UHSAS-1 additionally a thermodenuder (TD, T=300˚C) was installed so the flow first enters the switching (Hanbay) valve and either bypasses or passes through the TD before it enters the laminar flow element. Optical block schematic adopted from UHSAS User Manual.**



## 2.2 Modified flow system

Because the ground-based version of the UHSAS was not designed for operation on aircraft where pressure changes, flow system modifications are essential for airborne use. In the standard UHSAS configuration the aerosol sample flow is controlled and measured by a mass flow controller mounted on the exhaust side of the pump (Fig.1). If mass flow were maintained in flight, the volumetric flow rate would change inversely with air density, leading to changes in particle velocity through the laser beam and thus pulse width. A further issue is associated with transient sample flow response to pressure changes during aircraft altitude changes. Because the inlet nozzle restricts the sample flow entering the optical block, there is a time lag between any external pressure change and the pressure within the UHSAS optics block. This pressure disequilibrium changes the inlet flow to the optics block in a way that is dependent on the rate of pressure change and the fluid dynamics of the nozzle flow, which may vary with altitude because it depends upon Reynolds number ($R_e$). Because the particle number concentration is calculated from the measured count rate and sample flow rate, it is essential to account for these transient effects and directly measure the flow rate at the inlet. Finally, using a needle valve to control the split between the aerosol and sheath flows results in the sheath-aerosol flow ratio varying with changing pressure because pressure drop through the valve is also Reynolds-number-dependent and will vary with pressure, even at a constant volumetric flow rate.

Because of the above issues, the flow system of both UHSAS instruments was modified (Fig.1; Table S1). The modifications include installation of a laminar flow element with a differential pressure transducer to directly and precisely measure the time-varying sample volumetric flow rate at the optics block inlet, and replacement of the sheath flow valve with a volumetric flow controller (VFC) to directly monitor and control sheath flow. The Alicat mass flow controller on the exhaust side of the instrument, which is connected to an exhaust line near inlet pressure to control the exhaust flow, was switched to operate in volume flow control mode. The inlet laminar flow meter and differential pressure transducer were calibrated together over a flow range of 0-0.1 Lmin$^{-1}$ using a volumetric flow calibration standard (DryCal DC-Lite, Bios, Inc., Butler, NJ, USA). The modified UHSAS is operated at ~0.06 Lmin$^{-1}$ total inlet flow and 0.7 Lmin$^{-1}$ sheath flow. The original UHSAS LabView software was modified to accommodate these changes.

## 3. Laboratory performance

### 3.1 Aerosol generation method

The sizing performance of the UHSAS and the effects of particle composition and concentration were investigated in the laboratory (Fig.2). Particles were generated in two ways: 1) by using an atomizer to produce ammonium sulfate ($(NH_4)_2SO_4$), polystyrene latex (PSL) spheres, or di-2ethylhexyl (dioctyl) sebacate (DOS) particles (Table 1); or 2) from new particle formation and condensational growth from limonene ozonolysis products in a flow tube reactor.



### 3.1.1 Atomized aerosol and DMA

Particles were generated using an HPLC-grade water (or HPLC-grade isopropanol in the case of DOS) solution and a custom-built Collison-type atomizer (May, 1973). Atomized droplets were dried in a silica gel diffusion drier, charged by a Po$^{210}$ radioactive source and (except for the PSL) size-selected in a custom-built differential mobility analyzer (DMA) with a

recirculating sheath flow. The sizing uncertainty ($\sigma_S$) of the DMA was ±1.6 % estimated from the sum in quadrature (square root of the sum of squares) of the sheath flow ($\sigma_Q$), pressure ($\sigma_P$), temperature ($\sigma_T$) and voltage ($\sigma_V$) uncertainties as described in Eq.1.

$$\sigma_S = \sqrt{\sigma_Q{}^2 + \sigma_P{}^2 + \sigma_T{}^2 + \sigma_V{}^2} \tag{1}$$

A sizing bias to smaller diameters was identified when using NIST-traceable polystyrene latex (PSL) microspheres with

diameters between 0.07-0.4 µm (Thermo Scientific, Inc. Waltham, MA, US). This bias is estimated to be about 7 % at sizes below 0.07 µm and decreases to 1% for sizes above 0.13 µm. However, we believe that the actual bias is<7 % as these PSLs were checked against an independent DMA by P. Campuzano-Jost of the University of Colorado, and the results were similar, suggesting a surfactant coating on the smaller PSL sizes rather than a DMA sizing error. Still, these potential biases are propagated through to the aerosol surface and volume concentrations discussed below.

The calibration DMA operated at a 1:10 aerosol to sheath flow ratio and sheath flow rates of 3 to 5 L min$^{-1}$. The monodisperse aerosol flow exiting the DMA was diluted using particle-free air to match the flow rate of the instruments located downstream. The incoming particle free air was homogeneously mixed with calibration particles in a short section of turbulent (*Re*> 4000) flow and sampled by the two UHSAS instruments and a CPC (Model 3022A; TSI Inc., St. Paul, Minnesota, USA). The relative humidity *(RH)* of the aerosol flow was monitored by two *RH* sensors (Vaisala HMP60) installed in the DMA, one on the

sample flow exiting the DMA and the other on the sheath flow exiting the DMA column, and was typically <10 %. It was important to dry the atomized (NH$_4$)$_2$SO$_4$ aerosol prior to size classification to avoid sizing biases due to the uncontrolled evaporation of water in the DMA and UHSAS and refractive index effects in the UHSAS.

### 3.1.2 Flow tube reactor and DMA

A secondary organic aerosol (SOA) from limonene ozonolysis was generated in a borosilicate glass (Pyrex) flow tube reactor

as described in Williamson et al. (in preparation). Particles formed from limonene oxidation were size selected in a DMA as described above.



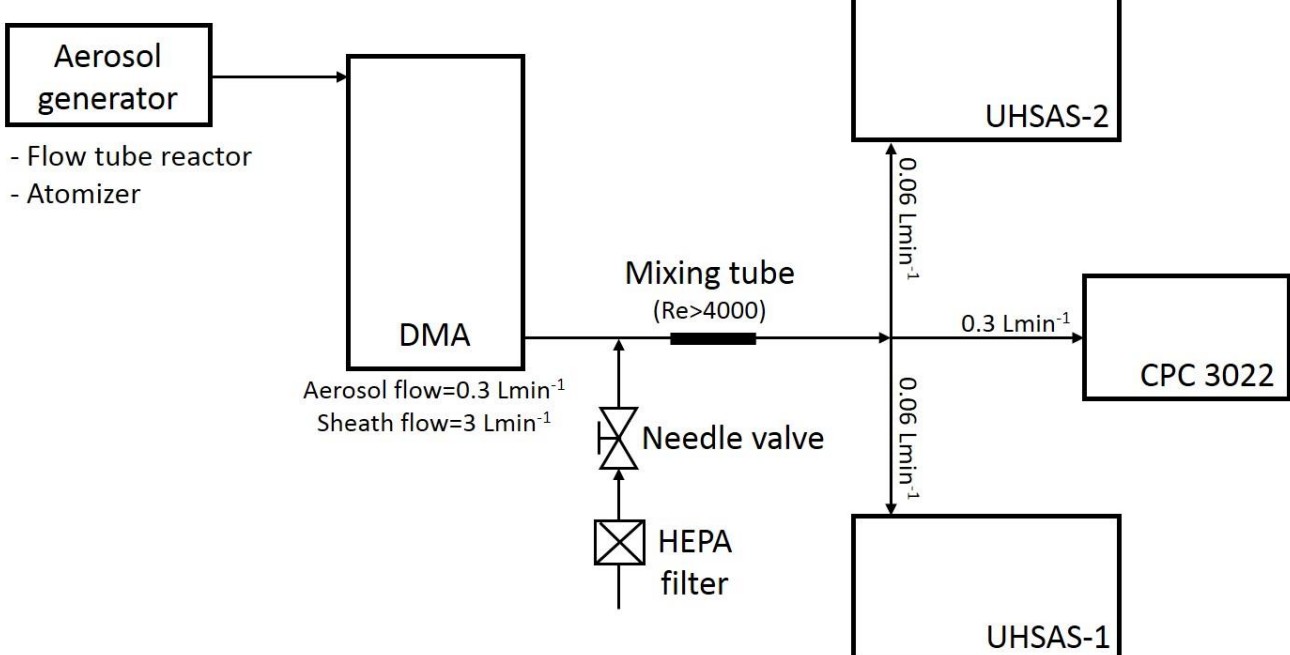

**Fig.2. A schematic diagram of the aerosol generation and measurement set-up at atmospheric pressure conditions. The calibration aerosol was generated either in a flow tube reactor or the atomizer. Apart from PSL, all atomized particles were sent through a diffusion drier to DMA for size selection, while PSL particles were delivered from the atomizer directly to both UHSAS instruments following dilution with dry air.**

**3.2 The effect of composition on particle sizing**

Particle sizing in the UHSAS is a function of the amount of light scattered onto the instrument's photodetectors. The quantity of scattered light however, is a function not only of size, but also of the composition-dependent aerosol refractive index (Bohren and Huffman, 1983). Particles of $(NH_4)_2SO_4$ were used to relate scattered light intensity to particle size, since their refractive index at 1054 nm (n=1.527; Hand and Kreidenweis, 2002) lies in the middle of the typical range of refractive indices for atmospheric particles composed of mixed sulfate salts and organic compounds. However, the composition of the atmospheric particles is not known *a priori*. The refractive index of organic aerosol in particular is not well constrained (Dick, 2007; Kanakidou et al., 2005). Kim and Paulson (2013) suggest values for refractive index (at $\lambda$=532 nm) for biogenic and anthropogenic secondary organic aerosol (SOA) of 1.44 and 1.55, respectively. To constrain the effects of particle refractive index on UHSAS sizing we investigated a range of nearly monodisperse calibration particles having different known refractive indices (Table 1, Fig.3), including $(NH_4)_2SO_4$, PSL microspheres, and DOS, as well as limonene oxidation products (with an unknown refractive index). For particles of $D_p< 0.6$ μm and a real refractive index (*n*) of 1.44-1.58, the diameter measured by





the UHSAS may vary by between +4/-10 % relative to the one based on $(NH_4)_2SO_4$. The propagation of this potential bias to reported aerosol surface and volume concentrations is discussed in Sect. 5.1.

Finally, we note that black carbon (BC) particles are mis-sized in the UHSAS. The optical cavity laser power is ~1 kW cm$^{-2}$, similar to that in the single particle soot photometer (SP2; Schwarz et al., 2010), and some limited laboratory studies we performed suggest that BC incandesces in the UHSAS. Because the number concentration of BC cores with volume-equivalent diameter (assuming void-free density of 1.8 g cm$^{-3}$) in the range 90-550 nm accounted for less than 5 % of the particle concentration in the same size range during the ATom-1 mission (except for the case of biomass burning plumes off the coast of Africa), mis-sizing due to BC is a minor effect in general in ATom. For cases of specific plumes from combustion sources in which BC is an abundant aerosol component this assumption should be re-evaluated.

Fig.3. Calibration particle diameter as a function of UHSAS-2 (not thermodenuded) bin number for particles composed of PSL, $(NH_4)_2SO_4$, DOS and limonene ozonolysis products. Uncertainties are shown for $(NH_4)_2SO_4$ and PSL but are often obscured by the symbols.




### 3.3 Particle detection efficiency

The detection efficiency, the ratio of concentration of particles of a given size measured by the sum of all bins of the non-thermodenuded UHSAS-2 to that measured by a TSI 3022A CPC, depends on the refractive index of the calibration particles used. Fig.4 presents the detection efficiency as a function of mobility equivalent diameter for $(NH_4)_2SO_4$ and DOS particles,

5    which varies due to the differing refractive indices of these compounds. The diameter uncertainties were calculated as described in Eq.1, and were corrected for the possible sizing bias observed using PSL standards. In a similar manner the uncertainties in the efficiency were calculated using the UHSAS and CPC uncertainties from the flow and pressure measurements and counting statistics.



**Fig.4. Detection efficiency of the non-thermodenuded UHSAS-2 instrument as a function of mobility equivalent diameter for $(NH_4)_2SO_4$ and DOS aerosol. Data are corrected for coincidence. Solid lines are fits presented to guide the eye.**



### 3.4 The effect of concentration on particle counting

The UHSAS sensitivity to particle concentration was quantified using atomized $(NH_4)_2SO_4$ particles with diameters >0.1 µm and concentrations between 1 and $10^4$ cm⁻³. All concentrations and flow rates presented in this paper are at STP conditions. The UHSAS exhibited a non-linear reduction in counting efficiency relative to the reference CPC at concentrations >1000 cm⁻

5 ³ due to particle coincidence in the optical sensing volume (Fig.5). Since the UHSAS software does not monitor and correct for coincidence effect, or live-time, while the CPC software does, we determined a phenomenological correction based on the observed counting efficiency as a function of count rate (Eq. 2):

$$N_{true} = \frac{N_{meas}}{1 - \tau * N_{meas} Q_{samp}},$$
(2)

where $N_{true}$ is the corrected number concentration (equal to the CPC concentration), $N_{meas}$ the measured number concentration,

10 and $Q_{samp}$ is the measured volumetric sample flow rate. Based on fitting the data in Fig.5 to Eq. 2, for the UHSAS-1, $\tau$=7.81x10⁻

⁵ s while for the UHSAS-2, $\tau$=5.36x10⁻⁵ s. These values represent the average particle pulse width for each instrument.

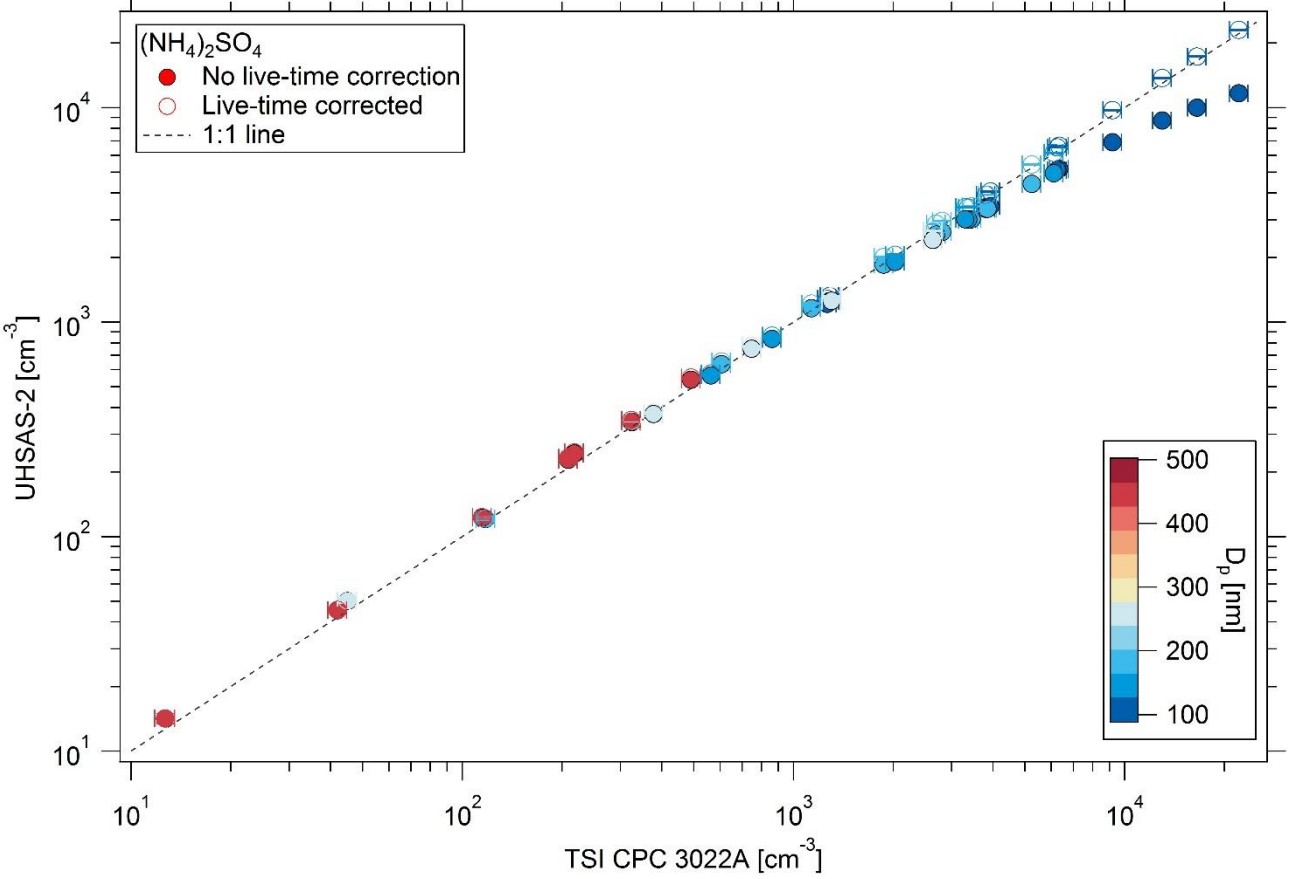

**Fig.5. Relationship between the UHSAS-2 and the CPC particle number concentration (cm⁻³) for nearly monodisperse $(NH_4)_2SO_4$ aerosol of various sizes (> 0.1 µm) at ambient pressure. Dashed line represents 1:1 correspondence line.**





## 3.5 The effect of pressure on particle sizing

Laboratory evaluation of the UHSAS operation at reduced pressure conditions is important for the interpretation and validation of the airborne data during the ATom flights. To investigate possible pressure dependencies, a needle valve and an external pump were used to reduce the instrument pressure. The flow passing through a needle valve downstream of the atomizer was

split into sample and bypass flows, the latter of which was connected to the pump. The exhausts of the UHSAS instruments were also connected to the bypass flow line to keep them at near-inlet pressure. A mixture of four PSL sizes was atomized and measured as instrument pressure was adjusted to as low as 250 hPa. The sizing of the UHSAS instruments showed no statistically significant pressure dependence (Fig.S1). The mean bin number and replicate standard deviation associated with each of the four PSL sizes at various pressure settings is 10.5 ±0.19, 24.5 ±0.24, 47.6 ±0.2 and 64.5 ±0.2 for the 81, 125, 240

and 400 nm PSL particles, respectively. Using the standard $(NH_4)_2SO_4$ calibration curve (Fig.3), which relates bin number to particle diameter, the equivalent relative standard deviations in diameter were ±0.6, 0.7, 0.5, and 0.7 % for the four diameters, respectively.

## 3.6 The effect of pressure on sample flow

Using the same set-up as described in the paragraph above, we investigated the effect of changing pressure on the sample flow. The aerosol volumetric flow rate showed a pressure dependence, decreasing from 60 cm$^3$ min$^{-1}$ at around 850 hPa to about 35 cm$^3$ min$^{-1}$ at 250 hPa (near the minimum pressure encountered during ATom). This flow reduction is caused by a small leak in the optics block downstream of the detection region. It was impractical to disassemble the complex optics assembly to find the source of this leak. Therefore, we directly measure the sample flow to account for this effect on concentration, and the leak

does not affect UHSAS sizing characteristics (Fig.S1).

## 4.0 Thermodenuder

A compact thermodenuder was designed and installed in UHSAS-1 to determine the number and volume fraction of volatile particles (Fig.6; Table S2). This measurement is used to identify particles that are formed from secondary products (e.g., sulfates, nitrates, and organics) from primary particles (e.g., soil dust and sea-salt; Clarke, 1991; Huffman et al., 2008).

Quantifying the volatile to non-volatile aerosol fraction during ATom may help improve understanding of the importance of secondary particles relative to sea-salt as CCN in the MBL, an area of active scientific inquiry (e.g., Bates et al., 2016; Quinn et al, in press).

## 4.1 Design

We constructed a custom thermodenuder based on the design principles outlined by Fierz et al. (2007), who improved denuder

performance by providing a heated adsorption section. This thermodenuder operates at a lower flow rate and is of a smaller




size compared to previous designs. An electric actuator (MDM-060DT, Hanbay Inc., Pointe-Claire, Quebec, Canada), driving a Swagelok valve (SS-43YF2) is used to automatically switch between sampling through the thermodenuder or bypassing it. The thermodenuder consists of a heated section (length, $L$=10.16 cm, inner diameter ($ID$)=0.48 cm) held at a fixed temperature ($T$=300˚C) followed by an adsorption section of same dimensions (Fig.6; Table S2). Both sections are housed in a stainless

steel tubing ($L$=30.48 cm, $OD$=1.27 cm) which contains an inner porous, perforated tube of the same length constructed from two pieces and manufactured using a metallic 3D printing technique, direct metal laser sintering (Xometry, Gaithersburg, MD, USA). This perforated tubing is wrapped with activated carbon fabric (Zorflex; 4.066 g). The outer tube passes through an aluminium housing which holds the tube and temperature sensor in place and is wrapped with a heating tape and fiberglass insulation material. Two fans installed in the outer casing of the heating section the entrance and exit sections of the

thermodenuder cool these sections of the outer tube. A thermal process controller monitors a resistance thermal detector (RTD) and controls the temperature of the aluminium bloc housing using a cylindrical cartridge heater. The temperature of the housing is maintained in flight at 300 ±0.5 ˚C. The residence time of the aerosol in the thermodenuder as well as the temperature profile in part determine thermodenuder performance. Fierz et al. (2007) developed simple guidelines for selecting an appropriate thermodenuder heated section length for a particular sample flow rate. Our thermodenuder meets these recommendations and

provides a residence time in the heated section between 1.59-3.7 seconds. We do not directly measure the thermal profile within the compact thermodenuder.

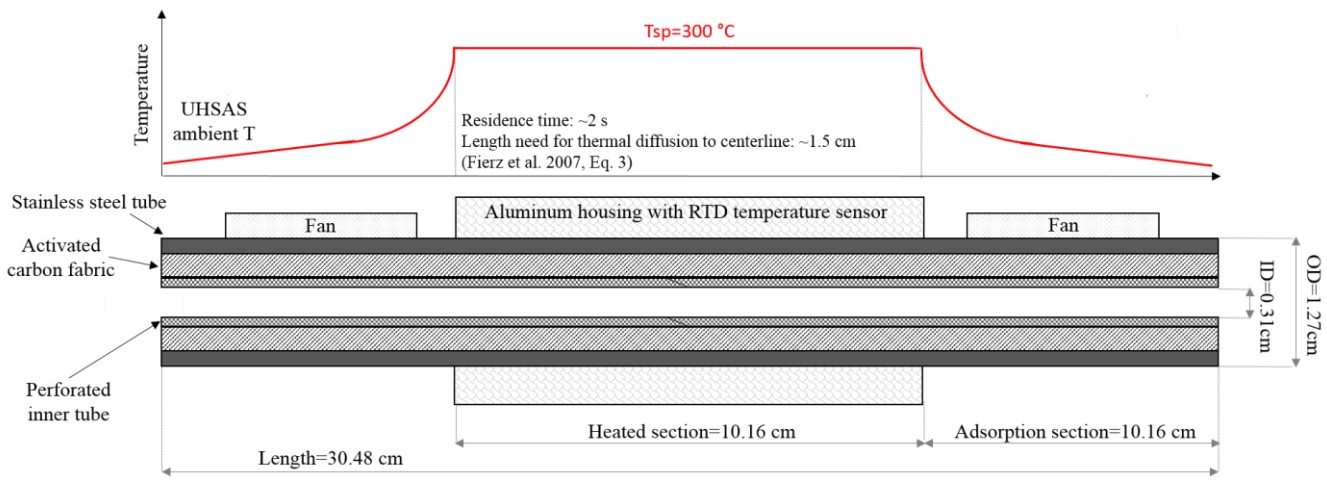

**Fig.6. Schematic cross-section of the thermodenuder and conceptual temperature profile. Temperature is measured at a single point with a platinum RTD sensor inside the aluminium housing around the heated section. The thermal diffusion length estimate assumes**

**standard pressure and temperature and typical flow in thermodenuder, small perturbations in temperature, and is use only for qualitative understanding of heat flow in the thermodenuder.**



## 4.2 Thermodenuder performance

Particle losses through the thermodenuder were determined relative to either a TSI 3022A CPC or the second UHSAS instrument. With the thermodenuder operating at room temperature, losses through the sample selection valve and heater plumbing were < 13 % for particles with $D_p$> 0.15 µm. With the heater on and the thermodenuder operating at 300 ˚C, losses

5 of non-volatile NaCl particles did not change significantly.

The efficiency of volatilizing particles in the thermodenuder was tested using DMA-size-selected particles from the generation of NaCl, $(NH_4)_2SO_4$, and limonene oxidation products at concentrations <1000 $cm^{-3}$. The UHSAS-1 alternated sampling between the thermodenuded and unheated sample lines every 2-3 minutes. The temperature of the thermodenuder was increased in steps from room temperature up to 310˚C and the fraction of particles exiting the thermodenuder (relative to the

10 unheated sample) at three different particle sizes was determined (Fig.7). Particles composed of $(NH_4)_2SO_4$ were most volatile, limonene oxidation products were less volatile, while NaCl was not volatile at the temperatures investigated. Smaller particles of $(NH_4)_2SO_4$ and limonene oxidation products volatilized at lower temperatures than larger particles of the same material, suggesting the particles were highly viscous, glassy, or solid. The effect of particle concentration on the performance was checked with particles generated from limonene oxidation products at 0.15 µm in diameter and concentrations of up to 11000

15 $cm^{-3}$. All particles at these concentrations were effectively volatilized, with no "break-through" effects observed. In no cases was there any evidence of recondensation of volatilized material to form new particles or to add material onto partially volatilized or non-volatile particles.





**Fig.7. Particle response to heating as a function of temperature of the thermodenuder, particle size and composition. Data normalized to the number measured at ambient temperature. Solid lines are used to guide the eye. The stability of the set temperature (dashed line) was within ±0.5 ˚C.**

## 5. Uncertainties

### 5.1 Uncertainties due to refractive index

Uncertainties in the aerosol volume and surface calculated from atmospheric dry size distributions depend on possible biases associated with the actual refractive index and shape of the particles vs. the calibration aerosol, as well as on random uncertainties associated with counting statistics, flow rate, pressure, and sizing precision and calibration accuracy. Since the ATom project focuses on the remote atmosphere where well-aged particles are expected to dominate the submicron aerosol (outside of sea-salt and dust cases), we did not investigate the effect of particle shape on sizing accuracy. Since the refractive index of organic compounds in the atmosphere is unknown but is likely bounded by our different calibration materials (e.g., (Kim and Paulson, 2013) we use the range of instrument response to the different calibration aerosols to estimate the likely effect of potential refractive index biases on aerosol volume and surface area derived from the UHSAS measurements.



As an example of the effect of these potential sizing biases on measured size distributions, we have selected a period of time from one of the ATom-2 flights (2017/02/10, Christchurch-Punta Arenas) when in the free troposphere (P~200-400 hPa). Using the range of instrument response curves for $(NH_4)_2SO_4$ ($n$=1.52), DOS ($n$=1.44) or PSL ($n$=1.58), the reasonable range of possible particle diameters associated with each UHSAS channel (bin) could vary by as much as +4/-10 % (as described in

5  Sect. 3.2). These diameter uncertainties propagate into aerosol volume and surface uncertainties +12.4/-27.5% and +8.4/-17.8 %, respectively as calculated from each 1s size distributions (Fig.8). Examples from this and other cases representative of conditions encountered during ATom flights are summarized in Table S3.

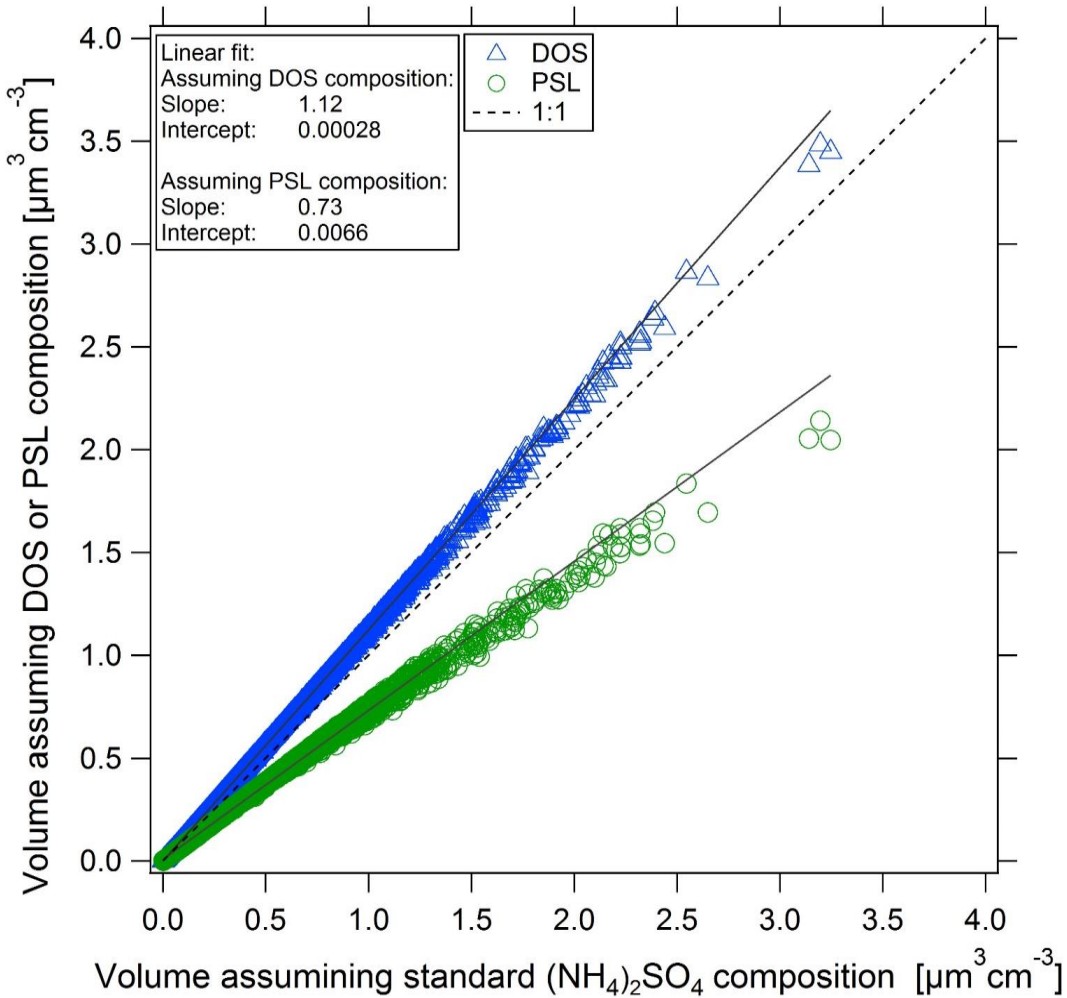

10  **Fig.8. Comparison of the calculated aerosol volume from the UHSAS-2 measured dry size distributions based on calibration particles with refractive indices between 1.44 and 1.58: $(NH_4)_2SO_4$ (n=1.52), DOS (n=1.44) and PSL (n=1.58). Flight data shown (2017/02/10) are from 1 s measurements. Solid lines represent a double-sided orthogonal distance regression linear fits.**





## 5.2 Uncertainties due to flow and pressure

Random uncertainties may arise from uncertainty in sample flow rates and uncertainty in the pressure measurement used to convert instrument concentrations to standard temperature and pressure (STP; 0°C and 1013 hPa). Uncertainty in the sample flow rate is ±0.86 % based on repeated calibrations of the sample flow meter over a range of 0-0.1 Lmin$^{-1}$ using a reference

calibration device (DryCal DC-Lite, Bios, Inc., Butler, NJ, USA). The uncertainty in the STP flow rate is the sum in quadrature of the flow calibration variation, the uncertainty of the DryCal flow calibration device (±0.25 %), the uncertainty in the differential pressure transducer reading (±0.25 %), and the uncertainty in the sample pressure (Eq. 1). The uncertainty of the measurement of the UHSAS-2 sample pressure at sea-level pressure is better than 0.38% when comparing to a reference pressure gauge. At <300 hPa, this pressure uncertainty was 3.8 % due to the lower accuracy of the pressure reference standard

used for lower pressures. The total propagated random uncertainty for the STP sample flow is <3.9 %.

## 5.3 Uncertainties due to counting statistics

Very low concentrations of accumulation-mode particles were often encountered in the free troposphere during the ATom mission. Uncertainties associated with resulting poor counting statistics at 1s resolution are reduced by averaging over longer time intervals. The uncertainty caused by the counting statistics was estimated for 1, 10 and 60s data averaging times using

various STP concentrations (20-440 cm$^{-3}$) representative of typical MBL and the upper FT conditions encountered (Table S3). As an example, uncertainties for STP concentrations of ~150 cm$^{-3}$ and ~30 cm$^{-3}$ as measured in the MBL for 1s acquisition intervals were ±8.7 and ±18 %, respectively. In the FT the uncertainties were much greater: ±14 and ±41 % for STP concentrations of ~440 cm$^{-3}$ and ~25 cm$^{-3}$, respectively. Actual instrument counting rates in the FT were much lower than for equivalent STP concentrations measured in the MBL because of lower air density.

## 5.4 Uncertainties due to instrument stability and calibration repeatability

Although careful calibrations undertaken using a DMA in the laboratory provide a precise assessment of UHSAS sizing characteristics, a method to validate the calibration stability of the UHSAS instruments in the field, where the DMA could not be carried, is critical. A solution of four PSL sizes (81, 125, 240 and 400 nm) in HPLC-grade water was atomized producing an aerosol with four distinct concentration peaks that could be measured by the UHSAS (Fig.9). The sizing channel associated

with each PSL diameter was determined by fitting a Gaussian curve to each peak in the size distribution histogram. The standard deviation of the identified peak bin was determined for a total of 84 calibrations taken before and after each flight, and at high altitude in-flight during test flights. The mean bin number and replicate standard deviation associated with each of the four PSL sizes is 10.8 ±0.4, 25.2 ±0.3, 47.4 ±0.2 and 64.6 ±0.2 for the 81, 125, 240 and 400 nm PSL particles, respectively. Using the standard $(NH_4)_2SO_4$ calibration curve (Fig.3), which relates bin number to particle diameter, for the UHSAS-2

instrument the equivalent precisions in diameter were ±1.2, 0.8, 0.7, and 0.7% for the four PSL sizes, respectively (Fig.9). Because the power in the optical cavity is sensitive to contamination of the optics, the UHSAS sizing calibration may shift



over time. This was observed during the middle of the Atom-1 mission in the UHSAS-1 when optical power dropped by 27 %. Because of the repeated calibration checks with the PSL particles, we were able to correct the observed size distribution with minimal errors despite the shift in calibration. Upon return to the laboratory, the instrument was recalibrated, then cleaned until laser power was restored and then calibrated again.

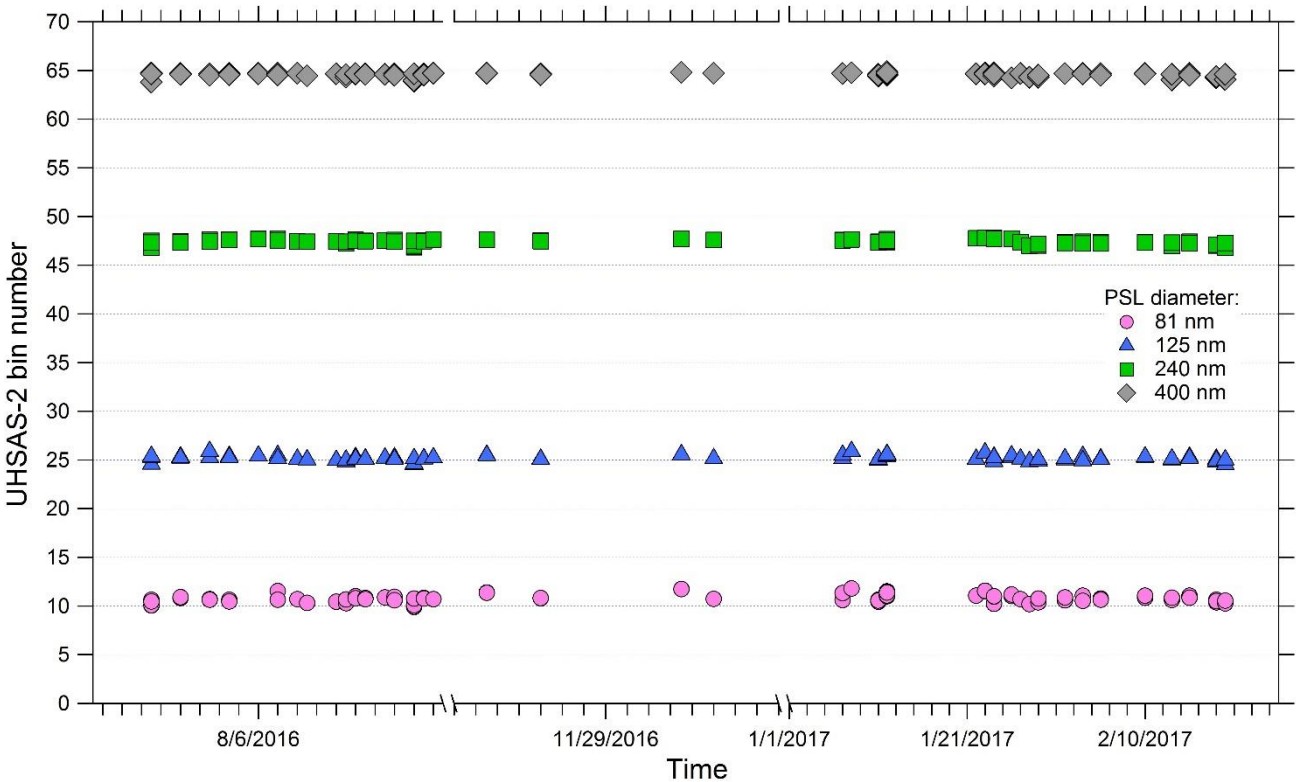

**Fig.9. Fitted peak bin number for four PSL size standards and as a function of time from July 2016-February 2017, showing calibration precision and the stability of the UHSAS-2 sizing during ATom-1 and -2.**

## 5.5 Total uncertainties

10 The total relative uncertainties for aerosol number concentration, surface and volume for cases of low and high particle number concentration measured in MBL and FT during ATom-2 mission are summarized in Table S3. The total uncertainty consists of random uncertainties due to the counting statistics, sample flow and pressure measurements, and possible systematic uncertainties due to sizing biases from the unknown refractive index of the atmospheric aerosol. The total uncertainty for aerosol number, surface and volume represents the sum in quadrature (Eq.1) of the random uncertainties plus the linear addition

15 of possible systematic sizing biases propagated through the surface and volume calculation.





We have not considered particle shape and homogeneity as a potential source of uncertainty. Given the laser wavelength of 1053 nm, and because most particles encountered in ATom were aged and likely only modestly aspherical, we do not expect shape sizing biases to be significant except for some larger sea salt and fresh dust particles.

## 6. In-flight performance

In this section, we describe the performance of the modified UHSAS instruments measuring dry aerosol size distributions, both directly sampled and thermodenuded, on the DC-8 aircraft during the ATom-1 (July-August 2016) and ATom-2 (January-February 2017) missions. Brock et al. (in preparation) more thoroughly describe the inlet and sampling configuration and provide comparisons between several aerosol instruments on the ATom payload. The measured internal UHSAS instrument pressures varied between ~1100 (due to ram pressure) and 225 hPa, which corresponded 0.15-13 km in altitude. The two UHSAS instruments sampled in parallel at 1 Hz downstream of a Nafion dryer that reduced sample RH to < 20 %. Periods of in-cloud measurement were excluded from the reported data due to aerosol sampling artefacts caused by droplet or ice crystal impacting the inlet, which produced spurious counts in the UHSAS instruments.

## 6.1 Consistency of aerosol number concentration, surface and volume measured by UHSAS-1 and UHSAS-2

During the ATom-1 deployment the thermodenuder on the UHSAS-1 instrument was not operated, allowing for direct comparison between the two UHSAS instruments. We compare number, surface, and volume concentrations over the diameter range from 0.1-0.9 µm to see if the measurements agree within the estimated uncertainties. We focus on the first five flights of ATom-1, between 2016 July 29-2016 August 8, before the laser power on the UHSAS-1 instrument shifted. The number, surface and volume concentrations were highly correlated between the two instruments ($r^2 > 0.98$), with slopes within 5 % of 1 (Fig.10). This agreement is well within the propagated uncertainties over the full dynamic range of 0-3000 cm$^{-3}$, 0-380 µm$^2$cm$^{-3}$, and 0-16 µm$^3$cm$^{-3}$ for number, surface area, and volume concentrations, respectively.

During ATom-2, when the UHSAS-1 was operated with the thermodenuder, the two UHSAS instruments could be compared by periodically switching the UHSAS-1 flow to bypass the thermodenuder when in MBL. During the non-thermodenuded sampling intervals, the agreement in concentration measured during first three flights (01/29-02/03/2017) over the Pacific was found to be between 0.97 ±0.011 and 1.04 ±0.01 (for 1 s data). The corresponding agreement for aerosol surface and volume concentration varied between 0.97-1.02 and 0.95-1.08, respectively.



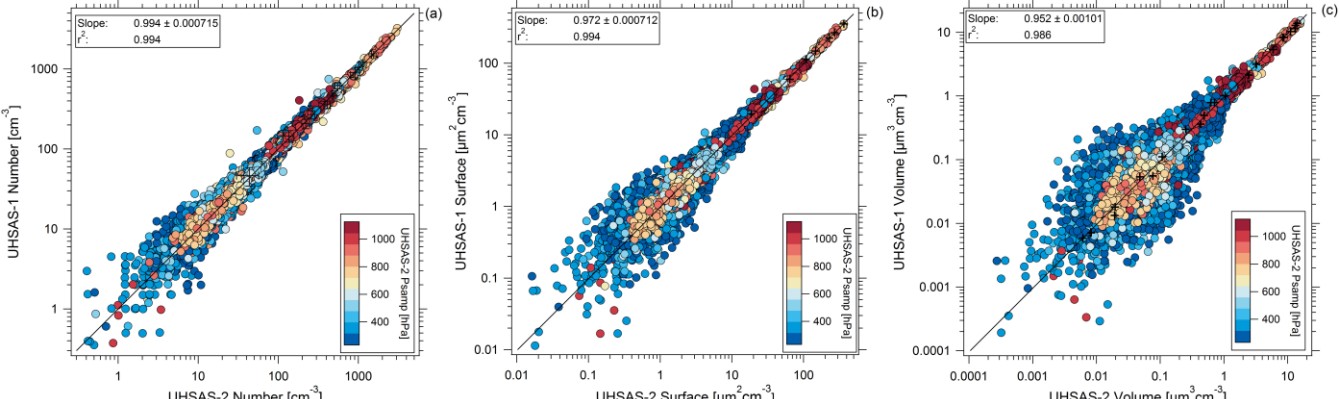

**Fig.10. Comparison between the UHSAS-1 and UHSAS-2 instruments on ATom-1 from 2016 July 29-2016 August 8 for (a) dry particle number, (b) surface area, and (c) and volume concentrations for diameters from 0.1-0.9 µm. Each point is a 10s average. The r² values indicated here refer to one-sided linear fit, while the solid lines represent double-sided orthogonal distance regression linear fits to non-transformed data. Estimated uncertainty is shown on a subset of points.**

## 6.2 Measurements of non-volatile aerosol fraction

The thermodenuded UHSAS was developed to help identify the fraction of particle number and volume (roughly proportional to mass) associated with primary particles such as sea-salt and dust as opposed to those that are produced by secondary processes (most particles composed of organic, nitrate, and sulfate species). Several measurements and modeling studies (Clarke and Kapustin, 2002; Korhonen et al., 2008; Mericanto et al., 2010; Quinn et al., 2017; Raes, 1995) suggest that secondary particles formed in the FT play an important role in governing CCN abundance in the MBL, despite the presence of sea-salt. It is possible that sea-salt may dominate aerosol mass in the MBL, but that CCN concentrations may be controlled by secondary processes, even those occurring in the FT above (Clarke et al. 2013; Raes, 1995; Twomey, 1977; Quinn and Bates, 2011).

To demonstrate the utility of the ATom UHSAS measurements for such investigations, we present examples of thermodenuded and non-thermodenuded aerosol number and volume size distributions for a single MBL case (measured for 50 s at 22˚N latitude over the central Pacific) and a single FT case (measured for 360 s at 3˚ N over the central Pacific) during ATom-2 on 2017 January 26 (Fig.11). In the MBL (Fig.11(a),(b)) volatile aerosol species dominate number concentrations, while non-volatile particles (presumably sea-salt) comprise ~54% of aerosol volume (or mass) for $D_p$ between 0.1 and 0.9 µm. The non-volatile (sea-salt) mode was largely >0.3 µm in diameter, clearly distinct from the smaller mode of volatile particles centered at ~0.15 µm volume mean diameter. Small amounts of non-volatile (sea-salt) mass extended down to diameters <0.1 µm, consistent with prior studies (Bates et al. 1998; Clarke et al., 1997; Mericanto et al., 2010; Middlebrook et al., 1998; Murphy et al. 1998; Quinn et al., 2017 in press).

The concentration of accumulation mode particles with $D_p$ between 0.1-0.9 µm in the clean air of the FT (Fig.11(c),(d)) was ~ 7 cm⁻³ and 94% of these were volatile. The peak modal diameter was smaller than could be detected by the UHSAS, implying

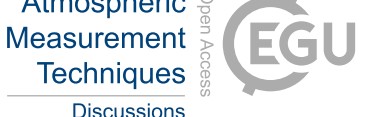



the dominance of the Aitken mode aerosol (0.012-0.06 µm). These particles were recently formed from gas phase precursors (Williamson et al., in preparation). In the FT, ~24 % of the particle volume was non-volatile, dominated by a few coarse-mode particles and uncertain due to poor counting statistics.

**Fig.11. Example of an averaged dry aerosol size distribution from UHSAS-1-TD and UHSAS-2 as sampled in the MBL (21.74° N, 1080 hPa) showing (a) number and (b) volume; and for a separate size distribution sampled in the FT (3.4° N, 293 hPa) showing (c) number and (d) volume. For particles with $D_p$ between 0.1 and 0.9 µm, isn the MBL case ((a), (b)) 94 % of the total number and 54 % of the total volume volatilized in the thermodenuder, while in the UT case ((c), (d)) 94 % of the number and 76 % of the volume volatilized.**

## 7. Summary and context

Two UHSAS instruments were modified, calibrated, tested in the laboratory, and operated during the first and second deployments of the ATom mission. The instruments are capable of continuous 1 s measurements of size-resolved particle



number concentration with high accuracy and precision over a diameter range of 0.063-1.0 µm from >1100 to 225 hPa, while simultaneously measuring particle volatility. Precision is limited by counting statistics, especially in the remote FT. The modified flow system of the UHSAS allowed direct monitoring of the sample flow rate and eliminated flow measurement issues associated with the pressure variations during aircraft altitude changes. The sizing of the UHSAS instruments showed

no statistically significant pressure dependence, crucial for consistent airborne sampling. Detailed calibrations with laboratory aerosols spanning a range of refractive indices (1.44-1.58) representative of the atmosphere allowed us to constrain the uncertainty associated with the unknown composition of the atmospheric aerosol. An equation to correct for particle coincidence was derived to improve the quantification of the counting accuracy at concentrations from ~1000 to >20,000 cm$^{-3}$. Two UHSAS instruments agreed in flight to within 5 % for integrated number, surface, and volume concentrations from sea

level to ~13 km altitude. We developed a compact thermodenuder for one of the UHSAS instruments, characterized its performance, and demonstrated its utility for quantifying size distribution of the nonvolatile fraction of the aerosol. Both modified UHSAS instruments worked well with no significant failures while flying on a DC-8 aircraft during the ATom missions.

The ATom observations taken with these instruments provided representative (non-targeted) measurements, across an

unprecedented latitude range over both ocean basins, of vertically resolved, size dependent aerosol properties that are related to radiative effects, to the ability of aerosols to act as CCN, and to the sources and abundance of primary vs. secondary particles in the MBL and FT. Hence, the size distribution data gathered by the UHSAS instruments over altitudes between ~0.2 and ~13 km will improve our understanding of global aerosol characteristics in the under-sampled regions of the atmosphere that closely resemble natural conditions minimally perturbed by pollution. These new measurements may be placed in the context of similar

data gathered over more than two decades by Clarke et al. (Clarke, 1991; Clarke et al., 2013; Clarke and Kapustin, 2002, 2010; Clarke et al., 1997; 1998), and others (e.g., Anderson et al., 1996) to help fill gaps in knowledge of aerosol properties, processes, sources, sinks, and aerosol-cloud-climate interactions.

The ATom measurements will continue in 2017 and 2018 with continued global measurements during the Northern Hemisphere fall and spring seasons. The past and future ATom measurements, placed in the context of chemical and

meteorological conditions and combined with size distribution measurements from 0.003-4.8 µm (Brock et al, in preparation; Williamson et al., in preparation), will help constrain model simulations of the processes that govern particle formation and their evolution in remote regions (Lee et al., 2013; Hamilton et al., 2014). Only if aerosol production mechanisms, sinks and transformations are understood can models accurately simulate global CCN distributions in the pre-industrial, modern, and future atmosphere, and the resulting effects on climate through aerosol-cloud interactions.

**Data availability**

Calibration and laboratory testing data are available upon request to the corresponding author. In-flight data are available at the ATom data archive: https://dx.doi.org/10.5067/Aircraft/ATom/TraceGas_Aerosol_Global_Distribution.



**Author contributions**

All authors contributed substantially to the work presented in this paper. A. Kupc and C.A. Brock modified the instruments. N. L.Wagner and C. A. Brock developed the thermodenuder. M. Richardson developed the software. A. Kupc designed, carried out experiments and analysed data. A. Kupc and C. Williamson calibrated instruments and collected data during ATom-1 and

-2 missions. A. Kupc prepared the manuscript with contributions from all co-authors.

**Acknowledgements**

The authors acknowledge support by NASA's Earth System Science Pathfinder Program under award NNH15AB12I and by NOAA's Health of the Atmosphere and Atmospheric Chemistry, Carbon Cycle, and Climate Programs. Agnieszka Kupc is supported by the Austrian Science Fund FWF's Erwin Schrodinger Fellowship J-3613. Droplet Measurement Technologies

kindly provided permission and source code to allow modification of the UHSAS control software. We would like to thank Joshua (Shuka) Schwarz and Joseph Katich for access to data and their helpful comments. We would also like to thank Bernadett Weinzierl, Maximilian Dollner, T. Paul Bui and Glenn S. Diskin for access to their preliminary data. Finally, we would like to thank David Fahey, Karl Froyd and Daniel M. Murphy for insightful discussions, and Pedro Campuzano-Jost and Jason C. Schroder for checking PSL standards.

**Disclaimer**

This publication's contents do not necessarily represent the official views of the respective granting agencies. The use or mention of commercial products or services does not represent an endorsement by the authors or by any agency.

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

**Table 1. Detection efficiency of UHSAS-1 and UHSAS-2**

| Particle | Real refractive index, n | Wavelength, λ (nm) | Reference | Dp$_{50}$ (nm) | |
|---|---|---|---|---|---|
| | | | | UHSAS-1 | UHSAS-2 |
| PSL | 1.58 | 780 | Yoo et al. (1996) | n/a | n/a |
| $(NH_4)_2SO_4$ | 1.527 | 1054 | Hand and Kreidenweis (2002) | 72.8 $^{+1.2}/_{-5.9}$ | 62.8 $^{+1.0}/_{-5.9}$ |
| DOS | 1.44 | 532 | Pettersson et al. (2004) | 75.9 $^{+1.2}/_{-6.0}$ | 68.2 $^{+1.1}/_{-5.9}$ |
| Limonene oxidation products | unknown | n/a | n/a | 78.9 $^{+1.3}/_{-6.0}$ | 69.7 $^{+1.1}/_{-5.9}$ |

