# Peer review of "Modification, Calibration, and Performance of the Ultra-High Sensitivity Aerosol Spectrometer for Particle Size Distribution and Volatility Measurements During the Atmospheric Tomography (ATom) Airborne Campaign"

_Atmospheric Measurement Techniques, 2017_

## Referee Comment (RC1) · Anonymous Referee #1 · 29 Sep 2017

This is a very thorough and well-written manuscript describing the evaluation and performance of the UHSAS instruments deployed during the ATom field campaign. The authors have investigated every aspect of the instrument performance and carefully quantified uncertainties. They have provided detailed descriptions of their methods and experimental details. The instruments performed with close agreement under similar operating conditions. The initial results with one thermodenuded instrument suggested a large fraction of the aerosols were secondary in nature. As this field campaign con-

[Figure]

tinues, these results should be very enlightening and help to further our understanding and constrain uncertainties regarding the ability of aerosols to act as CCN. My main concern/question is regarding the sensitivity of the instruments to particle refractive index and instrument calibration while operating in the thermodenuded mode. With the exception of sea salt which has a refractive index similar to ammonium sulfate or organic carbon (more volatile fractions), other non-volatile species such as dust or soot have a higher refractive index then was tested as part of the calibration (not to mention effects of complex refractive index). Are the authors concerned about different uncertainties between the thermodenuded and non-thermodenuded instrument when comparing the two measurements? I recommend publication after addressing very minor comments below.

Figure 3: Do the solid lines represent fits to the data? Please note in caption.

Page 18, line 25: I assume the "agreement" values correspond to slopes?

Page 19, line 25: Add "number" before "concentration"

Page 20, line 2: I'm not sure "coarse-mode" is typically used for particles less than 1 um?

Page 20, line 7, Figure 11 Caption: typo for "isn" before "the MBL case" and "UT" instead of "FT".

---

## Referee Comment (RC2) · Anonymous Referee #2 · 13 Oct 2017

The manuscript presents a detailed description of modifications to and the subsequent evaluation of two optical particle counters (UHSAS) for use during a multi-year aircraft campaign. It also describes and characterizes a thermal denuder for measuring the non-volatile fraction of aerosol. The subject is appropriate for AMT and the manuscript is well written and clear. I recommend publication once the following minor points have been addressed.

[Figure]

Section 3.1.1, line 10: Recommend changing "This bias" to be more specific, e.g., "The DMA sizing bias". Also, line 14 I believe the authors mean the biases are propagated to the aerosol and volume concentration uncertainties, correct? I assume no adjustment was made to the DMA diameters based on the PSL offset?

Section 3., lines 3-9: The section should also mention that the complex RI of BC will also affect the sizing in addition to LII impacts. I think it is also helpful to the reader to clarify how the incandescence of BC would affect UHSAS sizing (e.g., BC cores heat up and vaporize coatings and affect scattered light signals as the particle moves across the beam, other effects?).

Section 3.3: It appears the detection efficiency was only performed for UHSAS-2? Is there a reason for this? Even if results were similar it would be helpful to report diameters for 50% detection efficiency for both instruments for ammonium sulfate.

Section 4.2, lines 2-4: The particle losses through the TD are only reported down to 150 nm, but the both UHSAS systems measure down to about 70 nm. More information regarding particle losses through the TD should be provided between 70-150 nm. Also, will pressure in the TD affect the losses? I assume this varied over a similar range as described for the UHSAS? I do not think more experiments are needed, but some brief discussion of potential impacts would be useful.

Figure 6: What are the grey, circular "cloud" shapes on each side of the heated section at the top of the figure?

Section 5.3: Are sample flow rates changed during flights? A minor point, but it would be interesting to know if UHSAS response is affected by sample flow rate within a reasonable range. Assuming the residence time in the denuder could be maintained the count rate could be increased in the FT to improve the statistics and reduce uncertainties.

Section 6.1, line 17: I assume the reason to only compare 100-900 nm and not the

full UHSAS size range is to avoid slight variations in detection efficiency and satura-
tion? I am curious how well the instruments compare over the full range given by the
manufacturer.

---

## Referee Comment (RC3) · Anonymous Referee #3 · 13 Nov 2017

GENERAL COMMENT

The manuscript describes the results of a well-conducted study on the characteristics of two modified aerosol spectrometers (UHSAS) operated on the NASA DC-8 during the ATom airborne campaign. The implemented modifications concern mainly the stabilization of the sample flow at reduced and variable pressure levels which usually occur during airborne operation, and the introduction of a thermal denuder. The studies have been carefully designed and conducted, and the paper is clearly structured and well written. The topic fits well into the scope of AMT and the manuscript deserves publication after few minor revisions have been considered.

Minor revision are requested for these topics:

1. In Section 5, potential uncertainties in particle sizing from the unknown refractive index and the unknown impact of particle non-sphericity are not discussed in detail. For comparison, a detailed study on the impact of refractive index and shape uncertainties on particle size distributions determined by an an optical particle spectrometer is reported by Fiebig et al. (2002) for the PCASP which uses almost similar collection optics as the UHSAS. The authors may link their findings to these results to get an estimate of the excepted range of uncertainties.

2. Sections 3.5 and 3.6 may be combined since the only effect of pressure on particle sizing will arise from flow variations. A good example for the effect of an instable flow on the calibration of an optical particle counter is given by Bundke et al. (2015). The authors may refer to this instrument characterization study to compare their results.

3. In Section 3.1 the authors may add information n the size range of the produced aerosols. This would complete the information to the reader about the experiments performed in this study.

MINOR COMMENTS

Abstract: The abstract may be shortened to 250 − 300 words, e.g., the first sentence can be skipped and some details can be shifted to the text body.

Page 2, line 17: you may write: "to a size-proportional voltage pulse".

Page 2, line 25: I suggest rephrasing: "wished to dry the air sample and to install a thermodenuder used to distinguish non-volatile particles. These sample . . ."

Page 3, line 24: please modify "between $0.06 − 1$ $\mu$m in diameter".

Page 10, line 6: It should read: "life time".

REFERENCES

Bundke, U., Berg, M., Ibrahim, A., Tettich, F., Klaus, C., Franke, H., Fiebig, M., and Pet-zold, A.: The IAGOS-CORE aerosol package: Instrument design, operation and per-formance for continuous measurement aboard in-service aircraft, Tellus B, 67, 28339, doi: 10.3402/tellusb.v67.28339, 2015.

Fiebig, M., Petzold, A., Wandinger, U., Wendisch, M., Kiemle, C., Stifter, A., Ebert, M., Rother, T., and Leiterer, U.: Optical closure for an aerosol column: Method, accuracy, and inferable properties applied to a biomass-burning aerosol and its radiative forcing, 107, doi:10.1029/2000JD000192, doi: doi:10.1029/2000JD000192, 2002.

---

## Author Comment (AC1) · 15 Nov 2017

Authors' response to the review of the manuscript titled "Modification, Calibration, and Performance of the Ultra-High Sensitivity Aerosol Spectrometer for Particle Size Distribution and Volatility Measurements During the Atmospheric Tomography (ATom) Airborne Campaign"; submitted to AMT on August, 9, 2017

The authors would like to thank the reviewers of the manuscript for their careful and positive evaluations. Our responses are listed below in blue, while reviewers' comments are in black.

Apart from the minor changes suggested by reviewers we have updated data presented in Fig. 11. We have noticed that an incorrect dlogd values were used. The corresponding volatile and non-volatile fractions mentioned in Section 6.2 were also updated accordingly. These changed by ~1-2 % as compared to the initially reported values. This has not impacted integrated aerosol surface and volume concentration data reported here.
* * *
**Reviewer #1** (received and published: 29 September 2017)

This is a very thorough and well-written manuscript describing the evaluation and performance of the UHSAS instruments deployed during the ATom field campaign. The authors have investigated every aspect of the instrument performance and carefully quantified uncertainties. They have provided detailed descriptions of their methods and experimental details. The instruments performed with close agreement under similar operating conditions. The initial results with one thermodenuded instrument suggested a large fraction of the aerosols were secondary in nature. As this field campaign continues, these results should be very enlightening and help to further our understanding and constrain uncertainties regarding the ability of aerosols to act as CCN. My main concern/question is regarding the sensitivity of the instruments to particle refractive index and instrument calibration while operating in the thermodenuded mode. With the exception of sea salt which has a refractive index similar to ammonium sulfate or organic carbon (more volatile fractions), other non-volatile species such as dust or soot have a higher refractive index then was tested as part of the calibration (not to mention effects of complex refractive index). Are the authors concerned about different uncertainties between the thermodenuded and non-thermodenuded instrument when comparing the two measurements? I recommend publication after addressing very minor comments below.

We thank reviewer #2 for positive evaluation.

Because the refractive index of the atmospheric particles is not know a priori, we decided to use ammonium sulfate-based UHSAS sizing calibration and evaluated uncertainties with a reasonable range of scattering aerosols. Ammonium sulfate has a refractive index (m=1.527) that is representative for the remote marine aerosol sampled during ATom. We emphasize in the Section 3.2 that soot is not sampled quantitatively because of incandescence and/or refractive index. For dust we are concerned about both shape and refractive index. Uncertainties in both instruments for such cases must be evaluated on a case-by-case basis using best estimates of the refractive index based on other measurements, coupled with optical simulations of instrument response. We have added a remark to this effect in Section 3.2.

The fact that the two UHSAS instruments are measuring aerosol with different refractive indices is something we haven't considered quantitatively. As the reviewer points out, this is a particular problem above the MBL, where dust is likely the dominant component. Frankly, the counting statistics are sufficiently poor that this error is likely to dominate any uncertainties during comparisons between the two instruments in the free troposphere except in specific dust and dust + biomass burning cases. We have added

a remark cautioning about this sublety in Section 3.2, but believe this will have to be addressed on a case-by-case basis.

Figure 3: Do the solid lines represent fits to the data? Please note in caption.

Yes. Figure caption has been updated.

Page 18, line 25: I assume the "agreement" values correspond to slopes?

Yes. This sentence has been updated and now reads: "The corresponding slopes for aerosol surface and volume concentration varied between 0.97-1.02 and 0.95-1.08, respectively."

Page 19, line 25: Add "number" before "concentration"

Corrected.

Page 20, line 2: I'm not sure "coarse-mode" is typically used for particles less than 1 um?

The sentence reads now: "In the FT, ~3 % of the particle volume was non-volatile, dominated by a few particles with with $D_p$ >0.3 µm and uncertain due to poor counting statistics."

Page 20, line 7, Figure 11 Caption: typo for "isn" before "the MBL case" and "UT" instead of "FT".

Thanks for finding these subtle errors. We have fixed these two typos.
* * *
**Reviewer #2** (received and published: 13 October 2017)

The manuscript presents a detailed description of modifications to and the subsequent evaluation of two optical particle counters (UHSAS) for use during a multi-year aircraft campaign. It also describes and characterizes a thermal denuder for measuring the non-volatile fraction of aerosol. The subject is appropriate for AMT and the manuscript is well written and clear. I recommend publication once the following minor points have been addressed.

We thank reviewer #2 for positive evaluation.

Section 3.1.1, line 10: Recommend changing "This bias" to be more specific, e.g., "The DMA sizing bias". Also, line 14 I believe the authors mean the biases are propagated to the aerosol and volume concentration uncertainties, correct? I assume no adjustment was made to the DMA diameters based on the PSL offset?

Line 10: The sentence reads now: "This DMA sizing bias is estimated to be about 7 % at sizes below 0.07 µm and decreases to 1% for sizes above 0.13 µm."

Line 14: The sentence reads now: "Still, these potential biases are propagated through to the aerosol surface and volume concentration uncertainties discussed below."

Concerning the latter comment above. That is correct. No adjustment has been made to the DMA diameters based on the PSL offset. The sentence reads now: "No adjustments were made to the DMA diameters, but

the potential biases when compared to the PSL sizes are propagated through to the aerosol surface and volume concentration uncertainties discussed below."

Section 3., lines 3-9: The section should also mention that the complex RI of BC will also affect the sizing in addition to LII impacts. I think it is also helpful to the reader to clarify how the incandescence of BC would affect UHSAS sizing (e.g., BC cores heat up and vaporize coatings and affect scattered light signals as the particle moves across the beam, other effects?).

We agree. We have added the following text (page 8):

Line 3: "The refractive index of soil dust may exceed the range of real refractive indices considered here. In addition, dust can be both absorbing and aspherical. When dust is an important component of the atmospheric aerosol, uncertainties in both the denuded and thermodenuded UHSAS instruments should be evaluated on a case-by-case basis using best estimates of refractive index and shape based on other measurements, coupled with optical simulations of instrument response. Also, because the thermodenuded UHSAS instrument volatilizes non-refractory particles, the refractive indices in the aerosol measured by the two instruments will differ. This problem is probably minor in the MBL because sea-salt aerosol has a refractive index within the range of the calibrants. For the free troposphere, however, there may be substantial sizing biases between the two instruments that should be considered case by case using additional information on aerosol composition."

Line 13: "Even without incandescing, the complex refractive index of BC particles (n=2.26-1.26i at λ=1064 nm; Moteki et al. 2010) substantially alters UHSAS sizing compared with the calibration aerosol."

Reference added:

Moteki, N., Y. Kondo, and S. Nakamura, Method to measure refractive indices of small nonspherical particles: Application to black carbon particles, J. Aerosol Sci., 41, 513–521, 2010

Section 3.3: It appears the detection efficiency was only performed for UHSAS-2? Is there a reason for this? Even if results were similar it would be helpful to report diameters for 50% detection efficiency for both instruments for ammonium sulfate.

The detection efficiency was performed for both UHSAS-1 and -2 and results are reported in Table 1 (attached below for convenience). We did not include data for both instruments in Fig. 4 for clarity. We now refer specifically to Table 1 in Section 3.3 and comment that the detection efficiencies differ. The UHSAS-2 instrument is substantially older (serial #7) than the other, and has a lower laser power probably due to older, more contaminated optics.

We have added the following sentence: "Detection efficiencies for both UHSASs are provided in Table 1. The thermodenuded UHSAS begins detecting particles at a larger diameter than the other instrument."

**Table 1. Detection efficiency of UHSAS-1 and UHSAS-2**

| Particle | Real refractive index, n | Wavelength, λ (nm) | Reference | Dp50 (nm) | |
|---|---|---|---|---|---|
| | | | | UHSAS-1 | UHSAS-2 |
| PSL | 1.58 | 780 | Yoo et al. (1996) | n/a | n/a |
| $(NH_4)_2SO_4$ | 1.527 | 1054 | Hand and Kreidenweis (2002) | $72.8\ ^{+1.2}/_{-5.9}$ | $62.8\ ^{+1.0}/_{-5.9}$ |
| DOS | 1.44 | 532 | Pettersson et al. (2004) | $75.9\ ^{+1.2}/_{-6.0}$ | $68.2\ ^{+1.1}/_{-5.9}$ |
| Limonene oxidation products | unknown | n/a | n/a | $78.9\ ^{+1.3}/_{-6.0}$ | $69.7\ ^{+1.1}/_{-5.9}$ |

Section 4.2, lines 2-4: The particle losses through the TD are only reported down to 150 nm, but the both UHSAS systems measure down to about 70 nm. More information regarding particle losses through the TD should be provided between 70-150 nm. Also, will pressure in the TD affect the losses? I assume this varied over a similar range as described for the UHSAS? I do not think more experiments are needed, but some brief discussion of potential impacts would be useful.

We do not currently understand the loss mechanism in the TD. The loss is larger than we expect for flow through the unheated TD. Based on the particle size and residence time in the TD, diffusion to the walls is one possible loss mechanism and may be larger than expected due to the activated-carbon fabric in the TD. Another possible loss mechanism would be if a portion of the flow in the TD occurred in activated-carbon fabric region and the fabic acted as a filter leading to losses in the unheated TD. Further laboratory measurements are planned to understand these losses.

However, for this manuscript, without a clear mechanism for the losses, we cannot extrapolate to smaller size and lower pressures and can only report our measurements at 150 nm and 835 hPa. We have added a sentence to manuscript stating that the particle loss mechanisms in the TD are not clear: "The mechanism and size-dependence of this particle loss is currently unclear and requires further investigation."

Figure 6: What are the grey, circular "cloud" shapes on each side of the heated section at the top of the figure?

We are not sure what the reference is to. This is what we see on our screen and printouts.

[Figure]

**Fig.6. Schematic cross-section of the thermodenuder and conceptual temperature profile. Temperature is measured at a single point with a platinum RTD sensor inside the aluminium housing around the heated section. The thermal diffusion length estimate assumes standard pressure and temperature and typical flow in thermodenuder, small perturbations in temperature, and is use only for qualitative understanding of heat flow in the thermodenuder.**

Section 5.3: Are sample flow rates changed during flights? A minor point, but it would be interesting to know if UHSAS response is affected by sample flow rate within a reasonable range. Assuming the residence time in the denuder could be maintained the count rate could be increased in the FT to improve the statistics and reduce uncertainties.

The sample flow rate changes in flight. As described in section 3.6, the sample flow rate drops when the UHSAS operates at lowest pressures. This effect does not affect UHSAS sizing characteristics (Fig.S1). We agree that maintaining the flow rate constant in the free troposphere would improve the counting statistics, and reduce uncertainties.

Section 6.1, line 17: I assume the reason to only compare 100-900 nm and not the full UHSAS size range is to avoid slight variations in detection efficiency and saturation? I am curious how well the instruments compare over the full range given by the manufacturer.

Yes, we wanted to compare over a range where both instruments measure with 100% efficiency. As indicated in Table 1, the UHSAS-1 (thermodenuded instrument) is not as sensitive as the newer unit, and 100 nm is safely above the roll-off in detection efficiency to ensure a fair comparison. The diameter of 900 nm has been chosen as the upper limit of the UHSAS sizing calibration curve based on measured ammonium sulfate particles (Fig.3). Above 900 nm the calibration is based on fitted parameters rather than the actual measured calibration particles.
* * *
**Reviewer #3** (received and published: 13 November 2017)

GENERAL COMMENT
The manuscript describes the results of a well-conducted study on the characteristics of two modified aerosol spectrometers (UHSAS) operated on the NASA DC-8 during the ATom airborne campaign. The implemented modifications concern mainly the stabilization of the sample flow at reduced and variable pressure levels which usually occur during airborne operation, and the introduction of a thermal denuder. The studies have been carefully designed and conducted, and the paper is clearly structured and well written. The topic fits well into the scope of AMT and the manuscript deserves publication after few minor revisions have been considered.

We thank reviewer #3 for positive evaluation.

Minor revision are requested for these topics:
1. In Section 5, potential uncertainties in particle sizing from the unknown refractive index and the unknown impact of particle non-sphericity are not discussed in detail. For comparison, a detailed study on the impact of refractive index and shape uncertainties on particle size distributions determined by an an optical particle spectrometer is reported by Fiebig et al. (2002) for the PCASP which uses almost similar collection optics as the UHSAS. The authors may link their findings to these results to get an estimate of the excepted range of uncertainties.

We are not entirely sure we have fully understood reviewer's comment.

In Section 5.1 we stated that we did not investigate the effect of particle shape on UHSAS sizing accuracy. The ATom project focuses on the remote atmosphere where well-aged particles (assumed to be spherical) are expected to dominate the submicron aerosol (outside of sea-salt and dust cases). Examples of uncertainties representative of conditions encountered during ATom flights are summarized in Table S3.

The paper by Fiebig et al. (2002) focuses on biomass-burning aerosol properties and associated uncertainties. We feel that linking these two studies is not straight forward, therefore we decided not to include a reference to this paper here.

In section 3.2 we have added few sentences that might be relevant to the reviewer's comment above: "The refractive index of soil dust may exceed the range of real refractive indices considered here. In addition, dust can be both absorbing and aspherical. When dust is an important component of the atmospheric aerosol, uncertainties in both the denuded and thermodenuded UHSAS instruments should be evaluated on a case-by-case basis using best estimates of refractive index and shape based on other measurements, coupled with optical simulations of instrument response."

2. Sections 3.5 and 3.6 may be combined since the only effect of pressure on particle sizing will arise from flow variations. A good example for the effect of an instable flow on the calibration of an optical particle counter is given by Bundke et al. (2015). The authors may refer to this instrument characterization study to compare their results.

We agree and we have combined these two subsections and modified section 3.5 header to: "The effect of pressure on sample flow and particle sizing"

We mention in Section 3.5 that this reduction in sample flow at low pressure sampling conditions is caused by a small leak in the optics block downstream of the detection region. Therefore, we directly measure the sample flow to account for this effect on concentration. Further, we have shown that the leak does not affect UHSAS sizing characteristics (Fig.S1). This is a unique situation to our instrument therefore we feel the suggested citation is not needed here.

3. In Section 3.1 the authors may add information n the size range of the produced aerosols. This would complete the information to the reader about the experiments performed in this study.

Thank you for noticing. We have added a size range and the sentence reads now:

Page 5 line 27: "Particles with a range of diameters between 0.05 and 1 µm were generated in two ways: 1) by using an atomizer to produce ammonium sulfate (($NH_4)_2SO_4$), polystyrene latex (PSL) spheres, or di-2-ethylhexyl (dioctyl) sebacate (DOS) particles (Table 1); or 2) from new particle formation and condensational growth from limonene ozonolysis products in a flow tube reactor."

MINOR COMMENTS
Abstract: The abstract may be shortened to 250 – 300 words, e.g., the first sentence can be skipped and some details can be shifted to the text body.

Thank you for this suggestion, however we would like to keep the current version of the abstract.

Page 2, line 17: you may write: "to a size-proportional voltage pulse".

Corrected.

Page 2, line 25: I suggest rephrasing: "wished to dry the air sample and to install a thermodenuder used to distinguish non-volatile particles. These sample : : :"

Corrected.

Page 3, line 24: please modify "between 0.06 – 1 µm in diameter".

Corrected.

Page 10, line 6: It should read: "life time".

We think "live-time" is used there correctly.

[revised manuscript text omitted]